# Microbiology of Healthcare-Associated Infections: Results of a Fourth National Point Prevalence Survey in Serbia

**DOI:** 10.3390/antibiotics11091161

**Published:** 2022-08-28

**Authors:** Ivana Ćirković, Ljiljana Marković-Denić, Milica Bajčetić, Gorana Dragovac, Zorana Đorđević, Vesna Mioljević, Danijela Urošević, Vladimir Nikolić, Aleksa Despotović, Gordana Krtinić, Violeta Rakić, Ivana Janićijević, Vesna Šuljagić

**Affiliations:** 1Faculty of Medicine, University of Belgrade, 11000 Belgrade, Serbia; 2Institute of Microbiology and Immunology, 11000 Belgrade, Serbia; 3Institute of Epidemiology, 11000 Belgrade, Serbia; 4Institute of Pharmacology, Clinical Pharmacology and Toxicology, 11000 Belgrade, Serbia; 5Institute of Public Health of Vojvodina, 21000 Novi Sad, Serbia; 6Faculty of Medicine, University of Novi Sad, 21000 Novi Sad, Serbia; 7Department of Hospital Infection Control, University Clinical Centre of Kragujevac, 34000 Kragujevac, Serbia; 8Department of Hospital Epidemiology and Hygiene, University Clinical Centre of Serbia, 11000 Belgrade, Serbia; 9Ministry of Health, 11000 Belgrade, Serbia; 10General Hospital of Subotica, 24000 Subotica, Serbia; 11Institute of Public Health of Serbia, 11000 Belgrade, Serbia; 12Institute of Public Health of Niš, 18000 Nis, Serbia; 13Faculty of Medicine of Military Medical Academy, University of Defence, 11000 Belgrade, Serbia

**Keywords:** healthcare-associated infections, point prevalence survey, AMR, risk factors, Serbia

## Abstract

Millions of patients acquire healthcare-associated infections (HAIs) every year, putting them at risk for serious complications and prolonged hospitalization. Point prevalence surveys (PPS), guided by the European Centre for Disease Prevention and Control framework, are one of the primary methods by which countries in the European Union conduct surveillance of HAIs. Serbia, though not in the EU, implemented this approach in its national PPS. The microbiological and antimicrobial resistance (AMR) analyses comprised patients in 61 out of 65 hospitals included in the fourth PPS conducted in November 2017. A total of 515/12,380 (4.2%) of the adult patients included in the PPS had at least one HAI, with intensive care units carrying the highest prevalence of 15.9%. Urinary tract and surgical site infections were the most frequently identified types of HAIs (23.9% and 23.0%, respectively). *Enterobacterales* comprised almost half (47.0%) of all causative agents, most notably *Klebsiella* spp. (16.7%). AMR was very high for most pathogens—80.5% of nonfermentative Gram-negative bacilli were resistant to carbapenems whereas 62.9% of *Enterobacterales* were resistant to third generation cephalosporins. The calculated AMR index of 61% is one of the highest in Europe. Further efforts are needed to reduce the burden of HAIs in Serbia that carry very high resistance rates to antibiotics currently used in clinical practice.

## 1. Introduction

Healthcare-associated infections (HAIs) are a ubiquitous public health problem and pose significant challenges for patients and healthcare systems. As millions of patients develop HAIs every year in the European Union (EU) alone [1], strategies based on prevention and rigorous surveillance continue to be the primary forms of intervention [2]. They ensure implementation of better infection control practices, but also allow earlier identification of HAIs [3]. Effectively treating HAIs and reducing the risk of their development depends on these strategies, as antimicrobial resistance (AMR), now responsible for almost 5 million deaths each year, continues to grow as a global issue [4]. The most extreme forms of AMR are, in fact, seen in causative agents of HAI [5], which puts patients at risk from a plethora of complications, resulting in a prolonged length of stay and death.

To support the efforts in HAI surveillance, the European Centre for Disease Prevention and Control (ECDC) provides a comprehensive framework for conducting repeated point prevalence surveys (PPS) of HAIs and use of antimicrobial drugs in acute care hospitals in the EU, with the intent to determine the true burden of HAIs on a larger scale [6]. In addition to EU countries, members of the European Economic Area (EEA) have also started implementing this methodology. At the national level, Serbia conducted multiple PPS, but started using the ECDC framework for its previous national PPS conducted in 2010, joining the rest of the European countries in tackling this issue.

In the fourth and latest PPS conducted in 2017, Serbia used the same ECDC methodology to continue national surveillance of HAIs in acute care hospitals. Our goal, as part of the national PPS, was to estimate the prevalence of HAIs, investigate which types of HAIs are most frequently developing in our setting, and identify causative agents of HAIs coupled with AMR testing.

## 2. Results

A total of 12,380 adult patients were included in the PPS. The mean age was 60.4 years and a little over half of patients were female (6573, 53.1%). Patients < 40 years of age comprised 15.9% of our sample. The majority of surveyed patients were hospitalized in general medicine and surgical wards (45.0% and 39.5%, respectively). Patient distribution with respect to hospital size was reasonably balanced, with approximately a quarter of patients representing the four categories in our study (Table 1).

A total of 515/12,380 patients had at least one HAI. The total number of HAIs was 544 and 722 pathogens were identified. The prevalence of all HAIs was 4.4% (95%CI 4.0–4.7) and the prevalence of patients with at least one HAI was 4.2% (95% CI 3.8–4.5), with ICUs carrying the highest prevalence of 15.9% (95% CI 12.9–19.4), shown in Figure 1. When looking at the prevalence from the perspective of hospital size, the large hospitals (>1100 beds) showed a prevalence of 6.4% (95% CI 5.6–7.3), whereas the other hospital types showed relatively similar numbers (Figure 2).

The significantly higher proportion of patients with a HAI were males and older than those without a HAI. According to the McCabe score, patients with a HAI more fequently had a fatal underlying disease and were more frequently hospitalized in the ICUs and large hospitals. Urinary catheters, peripheral venous catheters, and mechanical ventilation were significantly more common in patients with hospital-acquired infections. Almost all of the patients with HAIs had previously used antibiotics (Table 1).

The distribution of main HAI types in our study is shown in Figure 3. The three most common types included urinary tract infections (UTIs) (n = 130, 23.9%), surgical site infections (SSIs) (n = 125, 23.0%) and pneumonia (PN) (n = 108, 19.9%). Other notable HAI types included gastrointestinal infections (12.7%), of which 85% were caused by *Clostridioides difficile* (*C. difficile*), and bloodstream infections (n = 66, 12.1%). Skin and soft tissue infections (3.9%), ear, nose, and throat infections (0.9%) and other types of infections (3.7%) were also identified. When the distribution of HAI types was analyzed in relation to hospital size, urinary tract infections were the most prevalent HAI in small- (<360 beds) and medium (360–574)-sized hospitals. SSIs made up almost a third (30.2%) of all HAIs identified in large hospitals (575–1100 beds), while UTIs, PN and SSIs were encountered in similar numbers in very large hospitals (>1100 beds).

Causative agents of registered HAIs, stratified across the most prevalent types of HAI, is shown in Table 2. Almost half of all HAIs were caused by *Enterobacterales* (47.1%), and were responsible for causing 70% of all UTIs, 52.8% of all skin and soft tissue infections, 46.3% of all pneumonias, and 44.2% of all bloodstream infections. Other notable groups include Gram-positive cocci and nonfermentative Gram-negative bacilli (19.9% and 19.7%, respectively).

Individual pathogens are outlined in Figure 4. *Klebsiella* spp. (16.7%) was most frequently isolated, followed by *Acinetobacter* spp. (15.2%), *C. difficile* (11.0%), and *P. aeruginosa* (10.5%).

Antimicrobial susceptibility testing of isolated pathogens is shown in Table 3. Resistance to carbapenems was very high in nonfermentative Gram-negative bacilli (80.5%), whereas *Enterobacterales* exhibited a lower resistance rate to carbapenems (35.9%), but high resistance to third generation cephalosporins (62.9%). No resistance to vancomycin was seen in *Staphylococcus* spp., but more than a third (35.5%) of staphylococci were resistant to oxacillin (methicillin) and 28.9% of *Enterococcus* spp. were resistant to glycopeptides. The AMR composite index was 61% (95% CI: 55–66%).

## 3. Discussion

This study marks the fourth national PPS on HAIs and the observed prevalence of patients with at least one HAI of 4.2% was below the EU average of 5.9% [7]. The presence of HAIs in our hospitals was lower compared to neighbouring countries such as Slovenia and Greece [8,9], where prevalence rates of 6.6% and 9.1% were reported, respectively, but they are also lower than countries such as Belgium (7.3%), and Finland (11%) [10,11]. Though similar rates were seen in more developed countries such as Austria (4.0%) and Switzerland (4.5%) [7,12], Serbia is yet to go below the 4.0% threshold to reach rates seen in the Netherlands (3.8%), Germany (3.6%) and Lithuania (2.9%) [7].

The introduction of PPS in our country, the first being in 1999 when prevalence of patients with at least one HAIs was 6.3% [13], has led to a significant change in how HAIs are monitored and reported. First, the locus of reporting responsibilities has shifted from regional public health institutions to the hospitals themselves, and today almost every hospital in Serbia has a dedicated team for reporting HAIs, consisting of a hospital epidemiologist and nurse. These hospitals are now responsible for submitting annual reports to the public health institutions that curate the data from around the country, forming a coherent system of national surveillance. Second, the results generated from these surveys led to the development of national guidelines for certain types of HAIs such as SSIs [14], but also procedures for hand hygiene, *C. difficile* infections and other infection control measures. Third, extensive education of healthcare staff through a range of workshops and training has led to an overall increase in awareness and importance of proper HAI monitoring and reporting. Accordingly, a decrease in the prevalence rate was observed. There were 3.1% in second national PPS conducted in 2005 [15], and 4.9% in third study in 2010 [16]. It is important to emphasize that apart from the second prevalence study, which was conducted in May, all the other studies, including the fourth, were carried out in November. Perhaps the seasonal variations in patients background and pathogens may partly explain the lowest HAI prevalence observed in 2005.

As expected, the highest prevalence rates with respect to type of department were seen in the ICU (15.9%). Although much higher than the ECDC reports from 2017 identifying a rate of 8.3% [17], the prevalence of HAIs in our ICUs seems to be lower compared to many countries individually across the EU/EEA. Slovenia reported the occurrence of HAIs as up to 30% in ICUs [8], while surveys from Poland found the prevalence to be close to 40% in intensive care [18].

When looking at the HAI types identified in our study, results point to UTIs (23.9%) and SSIs (23.0%) as the two most frequently registered. Though our findings differ from European reports that show respiratory tract infections as the leading HAI type, countries have reported varying results, including Switzerland, where SSIs comprised 30% of all HAIs, whereas data from Ukraine showed almost 60% of all HAIs to be SSIs [19]. In our setting, UTIs have been readily identified as the most common HAI type in two previous national PPS [14,15]. There is an urgent need to enhance preventive measures for these infections in our hospitals, especially the catheter-associated urinary tract infections, in accordance with current guidelines [20].

HAIs identified in our study were more frequent in patients with fatal underlying disease, presented by McCabe classification, and patients in very large hospitals who had any or multiple invasive devices [21,22,23]. The reliability of the McCabe score in the prediction of HAIs has already been proven [24,25]. It is well known that large hospitals, including university hospitals, take care of patients with more severe underlying diseases, and in whom many invasive procedures are performed.

The distribution of causative agents of HAIs in our study revealed significant differences compared to the majority of EU countries, where *Escherichia coli* has been established as the primary cause of HAIs [26]. In our PPS, *Klebsiella* spp. was the most frequently isolated pathogen, in 16.7% of cases. Additionally, *Acinetobacter* spp. and *P. aeruginosa* were the second (15.2%) and fourth (10.8%) most common pathogens. Such a pathogen landscape can only be compared to Greece and Romania, as both countries identified similar patterns of causative agents in their latest surveys [9,27].

This leads us to the more worrying finding of our study—antimicrobial resistance rates of isolated pathogens. Carbapenem resistance of 80.5% for non-fermentative Gram-negative bacilli (*Acinetobacter* spp. being the most prominent example) is very high, but this is not surprising based on previous results from Central Asian and European Surveillance of Antimicrobial Resistance (CAESAR) network, where very high resistance rates for *Acinetobacter* spp. have already been observed [28]. Similarly, resistance rates of *Enterobacterales* to carbapenems (35.9%) and third-generation cephalosporins (62.9%) suggest that they are no longer viable candidates for the empirical treatment of HAIs, and alternative therapeutic strategies should be sought out. Resistance of *Staphylococcus aureus* to oxacillin (35.5%), but not vancomycin (0%), was in line with reports showing very low rates of VRSA in Europe [29]. Vancomycin-resistant enterococci (VRE), on the other hand, were identified in 28.9% of cases, which was a finding of concern since Europe-wide resistance rates were identified at a much lower rate (7.3%) [30].

Finally, the most troubling observation of the fourth PPS was the AMR composite index of 61%. Apart from Romania that has reported a composite index of 68.9, we were the country with the highest AMR composite index in Europe. Reasons for such findings can be attributed to the significantly higher use of antibiotics compared to virtually all other European countries [31], often as a consequence of inappropriate use and self-medication [32]. Further efforts are needed to curb antimicrobial resistance in the community and in the hospital setting through education of the general public and the medical community.

The limitation of the study is a well-known limitation of the point prevalence survey, i.e., the observation of patients at one point in time. A further limitation of our study is that pediatric patients were not included. Although HAIs are a fundamental problem in all hospitalized patients, the most vulnerable are those at extremes of age. Therefore, the prevalence observed in our study may be underestimated because we only included adult patients over 18 years of age. Considering that there is a difference between children and adults in terms of epidemiology, causative agents, and infection sites of HAIs, we excluded pediatric patients from this study, as was the case in the first PPS study in Europe [33]. However, this study has several strengths. First, selection bias was avoided because all acute care hospitals in the country participated in the study. The ECDC HAI definitions and validated ECDC methodology for PPS were applied. Second, several theoretical and practical training for all data collectors were organized. Many data collectors also participated in previous studies. They worked in their hospitals on HAI prevention and control jobs. Third, the study was organized with the intent that the highly skilled epidemiologists from the regional public health institutes were coordinators for the hospitals in their region and supported in data collecting in accordance with the study protocol. It is particularly important to highlight the long-standing, well-established cooperation between hospitals and 25 public health institutes in our country, including the National Institute for Public Health. Recently conducted systematic review and meta-analysis suggest a sustained potential for the significant reduction in HAI rates in the range of 35–55% associated with multifaceted interventions irrespective of the economic setting, which encourages action planning [34].

## 4. Materials and Methods

### 4.1. Design and Study Setting

A cross-sectional point-prevalence study encompassing 65 Serbian hospitals (63 public, 2 private) was carried out in November 2017. The method of the study is presented in more detail in our previously published article [14] related to surgical patients. It was a sub-study within the national PPS designed and embedded to compare the characteristics of operated and non-operated patients, as well as to analyze the use of antibiotics in these two groups of patients. However, the data collected in the national PPS for all adult patients were used in this paper. To ensure comparability of obtained results with EU/EEA countries, the study design followed the ECDC case definitions and the point-prevalence survey protocol (version 5.3) [6]. This protocol was fully translated into Serbian to ensure maximum accuracy during data collection. Upon invitation by the Ministry of Health, hospital participation was voluntary, and several training sessions describing case definitions and the survey protocol were held for hospital staff, project coordinators, and other healthcare practitioners that were involved in the data collection process. Only inpatients admitted to the ward before 8 a.m. on the day of the survey, and not discharged from the ward during the conducting of the survey, were included. Patients in the emergency room, dialysis patients, patients in outpatient departments, and day patients (day cases) who did not stay overnight in the hospital were excluded. Hospitals varied in their bed size: <360 beds (n = 30, 49.2%); 360–574 beds (n = 16, 26.2%); 575–1100 (n = 10, 16.4%); and over 1101 beds (n = 5, 8.2%).

In order to determine microbiology profile and AMR of all HAIs, the data of 61 hospitals for adult acute-care were analyzed in this paper.

### 4.2. Data Collection

Data were collected in a single day, in one ward, with a maximum time frame of 2 weeks in one hospital, and within one month for the whole national survey. The first hospital started its survey on 26 October, and the last day of the survey at the hospital which was the last to start, was 26 November 2017.

Per the ECDC protocol, patient data were extracted from nursing and medical records and embedded into individual case report forms. The following information was obtained: demographics (age, gender), date of admission, type of ward, surgery since admission, McCabe score that determines the severity of the patient’s underlying condition, presence of invasive devices (central/peripheral vascular catheter, urinary catheter, intubation), and prior use of antimicrobials. In case antimicrobial use was confirmed, additional information was collected, such as route of administration, indication for use, start and duration of use, dosage, and potential change of use accompanied by its indcation.

In case an active HAI was suspected, the definition and criteria for HAIs were extracted from the ECDC case definitions—occurring after a minimum of >48 h, with the onset of symptoms on day 3 of hospitalization, day 1 being admission [27]. The following HAIs were extracted and classified: pneumonia, bloodstream infections (BSIs), urinary tract infections (UTIs), skin and soft tissue infections (SSI), and other. For each HAI, the site, the causative agent(s), the date of onset, and the results of antimicrobial susceptibility testing (AST) were gathered. Hospital-level data included information about the number of hospital beds and the number of intensive care units (ICUs) beds, as well as the type of hospital—secondary, tertiary healthcare level, or specialty hospital.

In addition to patient data, the case report form requires information about the type of hospital (secondary, tertiary, specialized care) and type of ward where the patient is hospitalized. In this paper we included patients older than 18 years from the following departments: ICU, surgical, medicine, gynecology, and mixed wards. Patients from geriatric, psychiatric and rehabilitation departments in participating hospitals with median hospitalization longer than 30 days were excluded. Due to significant differences between pediatric and adult populations in risk factors, causative agents, antimicrobial exposure, antimicrobial resistance, and consequently HAIs prevalence, we excluded patients younger than 18 years.

The antimicrobial resistance (AMR) composite index, used by the ECDC as a metric to establish the extent of AMR, was calculated using the ECDC definition: the percentage of resistant isolates to “first level” AMR parameters divided by the sum of isolates with confirmed AST [2]. First level markers include: methicillin resistance in *Staphylococcus aureus,* vancomycin resistance in *Enterococcus faecalis* and *Enterococcus faecium*, third-generation cephalosporin resistance in *Enterobacterales*, and carbapenem resistance in *Acinetobacter baumannii* and *Pseudomonas aeruginosa.*

### 4.3. Ethical Consideration

All data included in the analysis were previously anonymized. Furthermore, each hospital was assigned a code known only to the members of the overall study group. As a result, no additional approval was necessary for the study, in line with the ECDC PPS protocol guidance.

### 4.4. Statistical Analysis

The local infection-control team entered the data into ECDC’s HelicsWin.Net software that allows anonymous data entry and validation. Additional data analysis was performed using SPSS, version 17 (SPSS, Inc, Chicago, IL, USA). Results were expressed as the mean ± SD or as the proportion of the total number of patients. The prevalence of patients with at least one HAI was calculated as a percentage of patients with at least one HAI divided by a total number of patients. The prevalence of all HAI was calculated as a percent of all HAI divided by the number total number of patients. The χ^2^ test or Fischer exact test was used for categorical variables and relative risk, and their corresponding 95% confidence intervals (CI) were calculated. For parametric continuous variables, mean values were compared using the Student *t* test. For nonparametric continuous variables, the Mann–Whitney U test was used.

## 5. Conclusions

This fourth national PPS in Serbia identified a HAI prevalence lower than European average, however, highlighted the continuing burden that urinary tract infections and surgical site infections are placing on our acute-care hospitals. These findings, together with the increasing AMR in hospital settings, suggest that it is time to consider systematic interventions. It is this teamwork that should be used in the prevention and control of the spread of HAI by pathogens with a high resistance rate, which is one of the most worrying results of this study and requires urgent action. The results of this PPS from 2017 showed us which intervention must be performed in order to improve the quality of health care and reduce AMR. We should focus our energy on upgrading the established surveillance systems in hospitals, antibiotic stewardship, writing new and reviewing all existing guidelines for the prevention and control of specific HAIs and guidelines for reprocessing of equipment and environmental hygiene. One of the most important interventions should be the establishment of computerized tools designed to support diagnostic and therapeutic decision-making in order to improve clinical practice and care quality in our hospitals. Last but not least, it is important to broaden the knowledge of healthcare workers, as well as patients, on evidence-based infection control guidance and prevention successes.

## Figures and Tables

**Figure 1 antibiotics-11-01161-f001:**
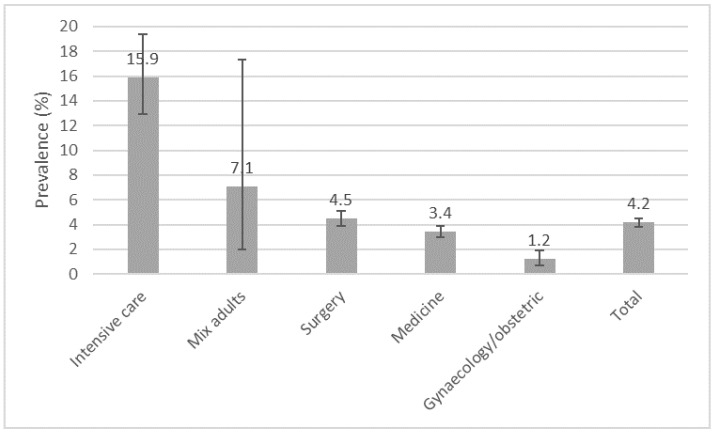
Prevalence of healthcare-associated infections (n = 515) by different hospital wards in the PPS in Serbia, 2017.

**Figure 2 antibiotics-11-01161-f002:**
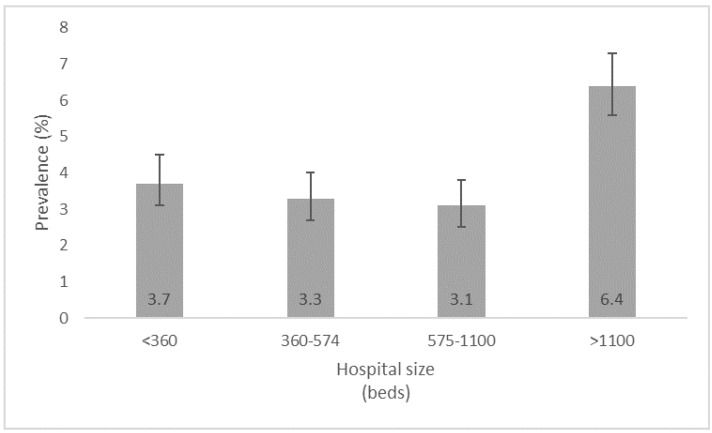
Prevalence of healthcare-associated infections across by hospital sizes in the PPS in Serbia, 2017.

**Figure 3 antibiotics-11-01161-f003:**
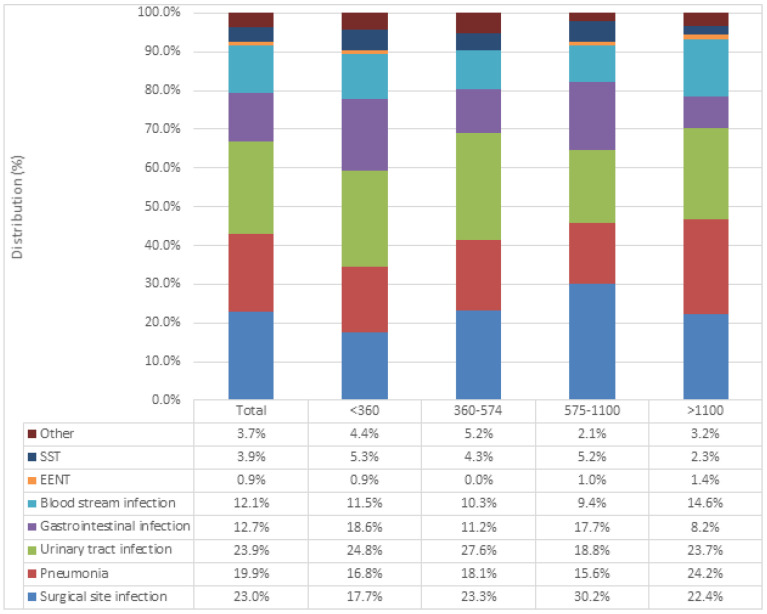
Main types of healthcare-associated infections, examined across different hospital sizes. SST: skin and soft tissue infection; EENT: ear, nose, and throat infection.

**Figure 4 antibiotics-11-01161-f004:**
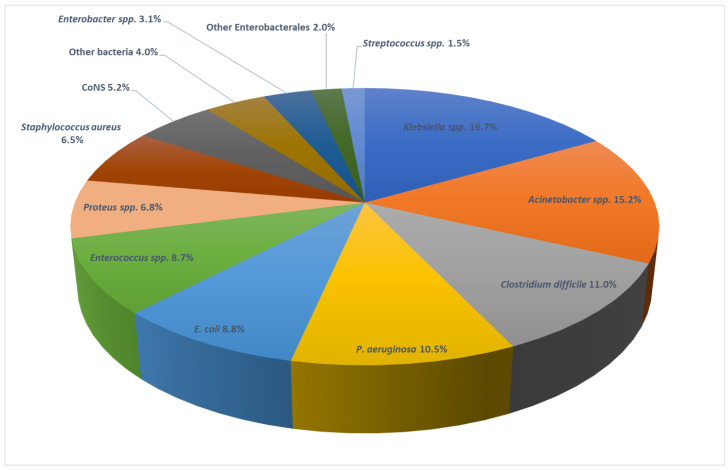
Causative agents of HAIs identified in the PPS in Serbia, 2017.

**Table 1 antibiotics-11-01161-t001:** Demographic and clinical characteristics of patients included in the PPS in Serbia, 2017.

	Total n (%)	Non-HAI n (%)	HAI n (%)	*p* Value
**Total**	12,380 (100.0)	11,865 (95.8)	515 (4.2)	
**Gender**				
Female	6573 (53.1)	6355 (53.6)	218 (42.3)	
Male	5807 (46.9)	5510 (46.4)	297 (57.7)	**<0.001**
**Age (mean** **± SD)**	60.4 ± 17.1	60.3 ± 17.1	63.8 ± 15.3	**<0.001**
<40 years	1963 (15.9)	1918 (16.2)	45 (8.7)	**<0.001**
40–59	2765 (22.3)	2650 (22.3)	115 (22.3)	0.998
60–79	6346 (51.3)	6056 (51.0)	290 (56.3)	**0.019**
>80	1306 (10.5)	1241 (10.5)	65 (12.6)	0.118
**Ward**				
Surgery	4896 (39.5)	4675 (39.4)	221 (42.9)	0.111
General Medicine	5565 (45.0)	5374 (45.3)	191 (37.1)	**<0.001**
Intensive care unit	521 (4.2)	438 (3.7)	83 (16.1)	**<0.001**
Gynecology	1342 (10.8)	1326 (11.2)	16 (3.1)	**<0.001**
Mix adults	56 (0.5)	52 (0.4)	4 (0.8)	0.263
**McCabe classification**				
Nonfatal	9714 (78.5)	9417 (79.4)	297 (57.7)	**<0.001**
Fatal within 5 years	654 (5.3)	581 (4.9)	73 (14.2)	**<0.001**
Fatal within 1 year	1663 (13.4)	1532 (12.9)	131 (25.4)	**<0.001**
Unknown	349 (2.8)	335 (2.8)	14 (2.7)	0.888
**Extrinsic factors**				
**Invasive devices**				
Urinary catheter	2823 (22.8)	2521 (21.2)	302 (58.6)	**<0.001**
Peripheral venous catheter	7871 (63.6)	7454 (62.8)	417 (81.0)	**<0.001**
Central venous catheter	631 (5.1)	485 (4.1)	146 (28.3)	**<0.001**
Mechanical ventilation	268 (2.2)	182 (1.5)	86 (16.7)	**<0.001**
**Exposure to intensive care**	521 (4.2)	438 (3.7)	83 (16.1)	**<0.001**
**Prior antibiotics therapy**	5242 (42.3)	4739 (39.9)	503 (97.7)	**<0.001**
**Hospital size**				
Small (<360 beds)	2909 (23.5)	2801 (23.6)	108 (21.0)	0.167
Medium (360–574)	3289 (26.6)	3180 (26.8)	109 (21.2)	**0.005**
Large (575–1100)	2928 (23.7)	2838 (23.9)	90 (17.5)	**0.001**
Very large (>1100)	3254 (26.3)	3046 (25.7)	208 (40.4)	**<0.001**

n: number of patients; SD: standard deviation; HAI: healthcare-associated infections; Significant *p* values are in bold.

**Table 2 antibiotics-11-01161-t002:** Distribution of main causative agents by healthcare-associated infection type in the PPS in Serbia, 2017.

	SSI		PN		UTI		GI		BSI		EENT		SST		Other		Total	
	n	%	n	%	n	%	n	%	n	%	n	%	n	%	n	%	n	%
**Gram-positive cocci**	66	31.4	7	5.2	26	16.0	0	0.0	36	37.9	2	40.0	6	16.7	1	5.9	144	19.9
**Gram-positive bacilli**	0	0.0	0	0.0	0	0.0	0	0.0	2	2.1	0	0.0	0	0.0	0	0.0	2	0.3
** *Enterobacterales* **	93	44.3	62	46.3	115	70.6	0	0.0	42	44.2	0	0.0	19	52.8	9	52.9	340	47.1
**Non-fermenting Gram-negative bacilli**	50	23.8	49	36.6	13	8.0	0	0.0	12	12.6	0	0.0	10	27.8	4	23.5	138	19.1
**Anaerobes**	0	0.0	0	0.0	0	0.0	62	89.9	1	1.1	0	0.0	0	0.0	0	0.0	63	8.7
**Fungi**	1	0.5	1	0.7	1	0.6	0	0.0	1	1.1	0	0.0	0	0.0	1	5.9	5	0.7
**Microorganism not identified or not found**	0	0.0	15	11.2	7	4.3	0	0.0	1	1.1	1	20.0	1	2.8	2	11.8	27	3.7
**Sterile examination**	0	0.0	0	0.0	0	0.0	2	2.9	0	0.0	0	0.0	0	0.0	0	0.0	2	0.3
**Result not available or missing**	0	0.0	0	0.0	1	0.6	5	7.2	0	0.0	2	40.0	0	0.0	0	0.0	1	0.1
**Total**	210	100	134	100	163	100	69	100	95	100.0	5	100	36	100	17	10	722	100

SSI: surgical site infection; PN: pneumonia; UTI: urinary tract infection; GI: gastrointestinal infection; BSI: bloodstream infection; EENT: ear, nose, and throat infection; SST skin and soft tissue infection.

**Table 3 antibiotics-11-01161-t003:** Antimicrobial susceptibility testing results for first-line antimicrobial resistance markers in the PPS in Serbia, 2017.

Antibiotic	Non-Fermenting Gram-Negative Bacilli	*Enterobacterales*	Gram-Positive Cocci		Total
	Resistant/tested (%)	Resistant/tested (%)	Resistant/tested (%)		Resistant/tested (%)
			*Staphylococcus* spp.	*Enterococcus* spp.	
CAR	103/128 (80.5%)	50/139 (35.9%)	N/A		158/282 (56.0%)
C3G	N/A	95/151 (62.9%)	N/A		105/168 (62.5%)
OXA	N/A	N/A	11/31 (35.5%)	0/1 (0%)	11/32 (34.4%)
GLY	N/A	N/A	0/27 (0%)	13/45 (28.9%)	13/72 (18.1%)

CAR: carbapenems (imipenem/meropenem); C3G—third generation cephalosporins (ceftriaxone/ceftazidime); OXA: oxacillin; GLY—glycopeptides (vancomycin); N/A—not applicable.

## Data Availability

Data supporting the results of this study are not publicly available but can be made available on request of the corresponding author.

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
