# Peer review of "Microbiology of Healthcare-Associated Infections: Results of a Fourth National Point Prevalence Survey in Serbia"

_antibiotics, 2022, doi:10.3390/antibiotics11091161_

Round 1

Reviewer 1 Report

The manuscript entitled “Microbiology and risk factors for healthcare-associated infections: results of a fourth national point prevalence survey in Serbia” estimated the prevalence of Healthcare-associated infections (HAIs), investigated which type of HAIs are most frequently developing in our setting, identified causative agents of HAIs coupled with antimicrobial resistance testing, and determined the risk factors for the development of HAIs. This is an important study in terms of addressing the prevention and control of the spread of HAI by pathogens with a high resistance rate. I have a minor suggestion: In conclusion, it should be elaborated that what kind of systemic interventions should be considered to deal with the scenario of HAIs and AMR. 

Author Response

Manuscript No. Antibiotics-1867177

Response to reviewers

Reviewers' comments:

Reviewer # 1

The manuscript entitled "Microbiology and risk factors for healthcare-associated infections: results of a fourth national point prevalence survey in Serbia" estimated the prevalence of Healthcare-associated infections (HAIs), investigated which type of HAIs are most frequently developing in our setting, identified causative agents of HAIs coupled with antimicrobial resistance testing, and determined the risk factors for the development of HAIs. This is an important study in terms of addressing the prevention and control of the spread of HAI by pathogens with a high resistance rate. I have a minor suggestion: In conclusion, it should be elaborated that what kind of systemic interventions should be considered to deal with the scenario of HAIs and AMR.

Response: Thank you for the suggestion.

We have added the following to the Conclusion: “The results of this PPS from 2017 showed us which intervention must be performed in order to improve the quality of health care and reduce AMR. We should focus our energy on upgrading the established surveillance system in hospitals, antibiotic stewardship, writing new and reviewing all existing guidelines for the prevention and control of specific HAIs and guidelines for reprocessing of equipment and environmental hygiene. One of the most important interventions should be the establishment of computerized tools designed to support diagnostic and therapeutic decision-making in order to improve clinical practice and care quality in our hospitals. And last, but not least, it is important to broaden the knowledge of healthcare workers, as well as patients, on evidence-based infection control guidance and prevention successes."

Additionally, we have added the following sentence to the Discussion: “Recently conducted systematic review and meta-analysis suggest a sustained potential for a significant reduction of HAI rates in the range of 35-55% associated with multifaceted interventions irrespective of the economic setting, which encourages action planning.” and, therefore, have added the reference No 34.

Reviewer 2 Report

Healthcare-associated infections (HAI) continue to be a global health problem. The authors report microbiology and risk factors for HAI in Serbia.

Abstract:

Line 29: HAI should be corrected to HAIs.

The conclusion should specify Serbia.

Introduction:

Well written and very informative.

Results:

The description of results for Table 1 is virtually the repetition of the results in the Table. Authors should summarize their findings using salient results in the Table.

I suggest lines 85-98 should be after lines 99-104.

Likewise, on line 121, ear-nose-and throat infections should be before other types of infections unless the authors are describing based on a particular trend (which be captured in their descriptions).

Table 3: N/A should be defined.

Table 4:  What informed selection of females, < 40 years, and surgery ward references in univariable and multivariable analyses?

Discussion

The authors have discussed their findings using available literature; however, I have the following concerns. They compared their results to that of Slovenia and Greece studies. These two studies and others cited have reported data for participants aged 0 to 102 and above for Slovenia and 0 to 75+ for Greece. Since it is known that HAI is higher in extremes of age, i.e. Neonates and very aged, I believe that the current study has a limitation that needs to be pointed out in the discussion.

Method

Why was the data of younger patients excluded from this study? Was it part of the ECDC Comprehensive for conducting repeated Point Prevalence Guideline?

The authors stated that they used first-level markers for determining the AMR Composite Index, and for Gram-positive bacteria, they used only S. aureus susceptibility to methicillin. Could the authors explain why they have not included S. haemolyticus (CoNS)? Although S. haemolyticus is considered an opportunistic pathogen, your data showed that catheter-related bloodstream infections are significant; moreover, you have a similar percentage of S. aureus (6.5%) and CoNS (5.2%) over 50% of your participant were 60 years and over. 

Author Response

Manuscript No. Antibiotics-1867177

Response to reviewers

Reviewer # 2

Reviewers' comments:

Healthcare-associated infections (HAI) continue to be a global health problem. The authors report microbiology and risk factors for HAI in Serbia.

Abstract:

Line 29: HAI should be corrected to HAIs.

            Response: We apologize for making this typo. We have changed HAI to HAIs.

The conclusion should specify Serbia.

            Response: Thank you for this suggestion. We have specified the place of study by adding "in Serbia."

Introduction:

Well written and very informative.

            Response: We thank the Reviewer for a positive comment.

Results:

The description of results for Table 1 is virtually the repetition of the results in the Table. Authors should summarize their findings using salient results in the Table.

            Response: Thank you for this invaluable comment. We have changed and reorganized that part of results and highlighted only the key results from Table 1.

I suggest lines 85-98 should be after lines 99-104.

            Response: Thank you for this comment, the lines 85-98 and 99-104 have been switched.

Likewise, on line 121, ear-nose-and throat infections should be before other types of infections unless the authors are describing based on a particular trend (which be captured in their descriptions)

            Response: Thank you for this comment. The order of infection types has been changed.

Table 3: N/A should be defined.

            Response: Thank you for pointing this out. We have added the meaning of N/A (not applicable) below the table.

Table 4: What informed selection of females, < 40 years, and surgery ward references in univariable and multivariable analyses?

            Response: Thank you for this suggestion. The reference category is based on the lowest prevalence of HAIs. To be consistent, we have changed the reference category for wards. The referent ward is Gynecology now, considering the smallest prevalence among all wards.

Discussion:

The authors have discussed their findings using available literature; however, I have the following concerns. They compared their results to that of Slovenia and Greece studies. These two studies and others cited have reported data for participants aged 0 to 102 and above for Slovenia and 0 to 75+ for Greece. Since it is known that HAI is higher in extremes of age, i.e. Neonates and very aged, I believe that the current study has a limitation that needs to be pointed out in the discussion.

Response: We thank the Reviewer for the invaluable comment.

We have added in the Discussion that one of the limitations of our study “is that pediatric patients were not included. Although HAIs are a fundamental problem in all hospitalized patients, the most vulnerable are those at extremes of age. Therefore, the prevalence observed in our study may be underestimated because we only included adult patients over 18 years of age. Considering that there is a difference between children and adults in terms of epidemiology, causative agents, and infection sites of HAIs, we excluded pediatric patients from this study, as was the case in the first PPS study in Europe”. Additionally, we have cited the reference No 33.

Method:

Why was the data of younger patients excluded from this study? Was it part of the ECDC Comprehensive for conducting repeated Point Prevalence Guideline?

            Response: Thank you for pointing this out. We decided to exclude younger patients from the study because evidence from the literature indicates differences between risk factors, causative agents, antimicrobial exposure, antimicrobial resistance and consequently HAIs prevalence. We have added explanation to the Methods.

The authors stated that they used first-level markers for determining the AMR Composite Index, and for Gram-positive bacteria, they used only S. aureus susceptibility to methicillin. Could the authors explain why they have not included S. haemolyticus (CoNS)? Although S. haemolyticus is considered an opportunistic pathogen, your data showed that catheter-related bloodstream infections are significant; moreover, you have a similar percentage of S. aureus (6.5%) and CoNS (5.2%) over 50% of your participant were 60 years and over.

            Response: We appreciate the comment and agree that Staphyloccocus haemolyticus is considered as an important catheter-related bloodstream infections pathogen.

In the present study, the composite index of AMR was calculated in accordance with ECDC PPS protocol (https://www.ecdc.europa.eu/en/publications-data/point-prevalence-survey-healthcare-associated-infections-and-antimicrobial-use-3), as it was used by other authors (Suetens C et al., Eurosurveillance, 2018; Silva A et al., Antibiotics, 2021; etc.).

Reviewer 3 Report

Dear Authors, 

The submitted manuscript entitled Microbiology and risk factors for healthcare-associated infections: results of a fourth national point prevalence survey in Serbia focuses on estimating the prevalence of hospital-acquired infections (HAI), identification of causative agents, and antimicrobial testing, as well as risk factors for the development of HAIs. The overall quality, writing, data presentation, and interpretation of results are good and comply with the journal standards, but my main concerns regard the already published results on this subject.

Major revisions:

1. What are the originality points of this manuscript in comparison with previously published results in BMC - Antimicrobials Resistance and Infections Control in 2021 (https://aricjournal.biomedcentral.com/articles/10.1186/s13756-021-00889-9)?

2. Is there a reason behind the different number of patients included in this manuscript in comparison with the number of patients included in the BMC article?

3. From the get-go, the previously published results should be mentioned and referenced, and a clear distinction regarding the originality should be made.  

These aspects need to be clarified and a rigorous verification should be performed as I am certain all authors are aware that redundancy in publications hampers researchers’ ability to find critical information quickly and efficiently. 

Author Response

Manuscript No. Antibiotics-1867177

Response to reviewers

Reviewers' comments:

Reviewer # 3

Dear Authors, 

The submitted manuscript entitled Microbiology and risk factors for healthcare-associated infections: results of a fourth national point prevalence survey in Serbia focuses on estimating the prevalence of hospital-acquired infections (HAI), identification of causative agents, and antimicrobial testing, as well as risk factors for the development of HAIs. The overall quality, writing, data presentation, and interpretation of results are good and comply with the journal standards, but my main concerns regard the already published results on this subject.

Major revisions:

1.What are the originality points of this manuscript in comparison with previously published results in BMC - Antimicrobials Resistance and Infections Control in 2021 (https://aricjournal.biomedcentral.com/articles/10.1186/s13756-021-00889-9)?

Response: We thank the Reviewer for raising this critical issue. We agree that the differences between the previously published article and the current manuscript were not clearly highlighted.

We agree that it made more sense to first publish this comprehensive paper on all HAIs in adults, followed by the article about surgical patients. However, driven by the need to innovate infection prevention recommendations in surgical patients, we first processed data for this group of participants.

We appreciate the consideration given by the Reviewer to our data set and manuscript, and we understand the concern regarding the originality, with which we entirely agree. Therefore, we have decided to omit the risk factors for HAIs. In this way, there is no overlap between the results of our previous article and the current one.

In the revised version of the Manuscript, the parts about risk factors are deleted in the Title, Objectives, Material and Methods, Results and Discussion sections.

To respond to the Reviewer's concern, we comparatively stated the key differences between our two articles:

  1. a) Objectives:

- In the previous article, the objective was to estimate "the prevalence of HAI and AMU in patients who recently had surgery in Serbian acute-care hospitals. Furthermore, risk profile, HAI rates, and AMU in this population were compared to the ones of non-surgical patients".

- In this manuscript, "Our goal, as part of the national PPS, was to estimate the prevalence of HAIs, investigate which types of HAIs are most frequently developing in our setting, and identify causative agents of HAIs coupled with AMR testing. In addition, we sought to determine risk factors for the development of HAIs".

The titles of the articles referred to the objectives of the articles: "A nationwide assessment of the burden of healthcare-associated infections and antimicrobial use among surgical patients: results from Serbian point prevalence survey, 2017" and "Microbiology of healthcare-associated infections: results of a fourth national point prevalence survey in Serbia"

  1. b) Material and methods:

In the Material and Method section in current manuscript, it was already highlighted that "to determine microbiology profile and AMR, the data from 61 hospitals for adult acute-care were analyzed". We have added a preceding sentence to that: "The method of the study is presented in more detail in our previously published article [14] related to surgical patients. It was a sub-study within the national PPS designed and embedded to compare the characteristics of operated and non-operated patients, as well as to analyze the use of antibiotics in these two groups of patients. However, the data collected in the national PPS for all adult patients were used in this paper.”

  1. c) Results:

- In the previous article, we compared the characteristics of surgical and non-surgical patients in Table 1. Additionally, the difference in HAIs prevalence between surgical and non-surgical patients across different hospital sizes was presented in Figure 2. Furthermore, the HAIs prevalence, the prevalence of surgical site infections and the antibiotic consumption only among patients according to the type of surgical procedure were included in Table 2.

- In this manuscript, we observe the characteristics of all adult patients included in the study and the prevalence of HAIs among all patients across different hospital wards and hospital sizes. Moreover, the main causative agents and their distribution by HAIs are shown, as well as their resistance to antimicrobial agents.

Regarding the amount of data we collected in the PPS, we believe two papers can be prepared, as was the case for ECDC PPS 2016-2017 participating countries or individual European countries.

We want to thank Reviewer #3 for her/his helpful comments on our manuscript.

2.Is there a reason behind the different number of patients included in this manuscript in comparison with the number of patients included in the BMC article?

Response: We thank the Reviewer for the invaluable comment. The difference between our previous article and this one is that we excluded 98 patients from geriatric, psychiatric, and rehabilitation departments in participating hospitals. Given that only a few hospitals have smaller wards for these patients and the different characteristics of HAIs in those wards, we excluded them from our current study. In Serbia, there are special hospitals for geriatric and psychiatric patients and rehabilitation, but they do not belong to acute care hospitals.

In the Method section, we have already stated that patients from these departments were excluded. We have modified the existing sentence in Data Collection section of Material and Methods as followed: "Patients from geriatric, psychiatric and rehabilitation departments in participating hospitals with median hospitalization longer than 30 days were excluded").

  1. From the get-go, the previously published results should be mentioned and referenced, and a clear distinction regarding the originality should be made.  

These aspects need to be clarified and a rigorous verification should be performed as I am certain all authors are aware that redundancy in publications hampers researchers' ability to find critical information quickly and efficiently. 

Response: We appreciate the Reviewer's suggestion. We would first like to point out that all authors are aware that any published article should bear the seal of originality. The detailed differences between our previously published article and this manuscript have already been explained in our answer to the first question. Here we listed the modifications that have been made to the manuscript in order to make the differences more noticeable.

Our previous article was already cited (reference 14). To clarify the similarities and differences between the Methods of our two papers, we have changed the order of the sentences in the section Design and Study Setting of Materials ad Methods. Additionally, we pointed out that the previous article referred only to surgical patients.

In order to increase the originality of the article, risk factors were omitted in the revised version of the manuscript (in the Title, Objectives, Material and Methods, and Results).

Round 2

Reviewer 3 Report

Dear Authors, 

Thank you for addressing the mentioned issues with elegance.

The manuscript is improved significantly and it is suitable for publication.